# Exploring the Role of Desmoplastic Physical Stroma in Pancreatic Cancer Progression Using a Three-Dimensional Collagen Matrix Model

**DOI:** 10.3390/bioengineering10121437

**Published:** 2023-12-18

**Authors:** Xiaoyu Song, Yuma Nihashi, Masamichi Yamamoto, Daiki Setoyama, Yuya Kunisaki, Yasuyuki S. Kida

**Affiliations:** 1Tsukuba Life Science Innovation Program (T-LSI), School of Comprehensive Human Sciences, University of Tsukuba, Tsukuba 305-8572, Japan; son.xiaoyu93@aist.go.jp; 2Cellular and Molecular Biotechnology Research Institute, National Institute of Advanced Industrial Science and Technology (AIST), Tsukuba 305-8565, Japan; y-nihashi@aist.go.jp; 3Department of Research Promotion and Management, National Cerebral and Cardiovascular Center, Kishibe-Shimmachi, Suita 564-8565, Japan; myamamoto@ncvc.go.jp; 4Department of Clinical Chemistry and Laboratory Medicine, Kyushu University Hospital, Fukuoka 812-8582, Japan; setoyama.daiki.753@m.kyushu-u.ac.jp; 5Department of Clinical Chemistry and Laboratory Medicine, Graduate School of Medical Sciences, Kyushu University, Fukuoka 812-8582, Japan; kunisaki.yuya.519@m.kyushu-u.ac.jp; 6School of Integrative & Global Majors, University of Tsukuba, Tsukuba 305-8572, Japan

**Keywords:** type I collagen, pancreatic ductal adenocarcinoma (PDAC), 3D matrix, tumor microenvironment

## Abstract

Pancreatic ductal adenocarcinoma (PDAC) is a refractory tumor with a poor prognosis, and its complex microenvironment is characterized by a fibrous interstitial matrix surrounding PDAC cells. Type I collagen is a major component of this interstitial matrix. Abundant type I collagen promotes its deposition and cross-linking to form a rigid and dense physical barrier, which limits drug penetration and immune cell infiltration and provides drug resistance and metabolic adaptations. In this study, to identify the physical effect of the stroma, type I collagen was used as a 3D matrix to culture Capan-1 cells and generate a 3D PDAC model. Using transcriptome analysis, a link between type I collagen-induced physical effects and the promotion of Capan-1 cell proliferation and migration was determined. Moreover, metabolomic analysis revealed that the physical effect caused a shift in metabolism toward a glycolytic phenotype. In particular, the high expression of proline in the metabolites suggests the ability to maintain Capan-1 cell proliferation under hypoxic and nutrient-depleted conditions. In conclusion, we identified type I collagen-induced physical effects in promoting Capan-1 cells, which cause PDAC progression, providing support for the role of dense stroma in the PDAC microenvironment and identifying a fundamental method for modeling the complex PDAC microenvironment.

## 1. Introduction

Pancreatic ductal adenocarcinoma (PDAC) is the most common type of pancreatic cancer, accounting for >85% of all diagnoses [1]. PDAC is a refractory and aggressive malignancy with a dense fibrotic stroma that is associated with poor prognosis [2,3], reflected in a 5-year survival rate of <10% [4]. This dense stroma presents as a fibrous interstitial matrix surrounding PDAC cells, which limits drug penetration and immune cell infiltration and provides drug resistance and metabolic adaptations to PDAC [5].

Collagen, primarily types I, III, IV, V, and XV, serve as key components of the extracellular matrix (ECM) in PDAC [6]. Of these, type I collagen is the most abundant and is present three times higher than that in the normal pancreas [7,8,9,10]. It tends to deposit and crosslink excessively in the matrix, forming a stiff, physical barrier. Moreover, type I collagen is responsible for most desmoplastic reactions, disrupting the structure of the basement membrane and inducing PDAC cells to increase interstitial collagen levels, resulting in the deterioration of PDAC [11]. Type I collagen can interact with integrin α2β1 expressed on the surface of PDAC cells, promoting their proliferation and migration [6,10]. This process results in the loss of the E-cadherin complex and cell–cell adhesion, an important step in metastasis and epithelial-to-mesenchymal transition (EMT) [12]. Thus, the composition of the ECM plays a pivotal role in driving EMT, and a type I collagen-rich microenvironment favors a more aggressive PDAC phenotype.

Collagen-based 3D cultures have profound epigenetic effects on PDAC cells [11]. When PDAC cells are cultured in a 3D collagen matrix, the expression of histone acetyltransferases p300, P300/CBP-associated factor, and GCN5 are increased [13], which is accompanied by an elevated level of high-mobility group A2, an epigenetic regulator of proliferation, apoptosis, and DNA repair [14]. In addition, collagen-based 3D cultures promote angiogenesis and the secretion of growth factors, such as epidermal growth factor (EGF), matrix metalloproteinase 9 (MMP9), and vascular endothelial growth factor (VEGF) [15]. However, our understanding of epigenetic modifications remains incomplete, highlighting the need for further studies surrounding the impact of collagen-based 3D cultures on PDAC cells.

In this study, a type I collagen-based 3D PDAC model containing Capan-1 cells (a PDAC cell line) was constructed to better understand the impact of collagen on the progression of PDAC tumor microenvironments. This 3D culture model is more informative than 2D culture systems for exploring the spatial structure of tumors, growth patterns, and cell–cell interactions [16]. Remarkably, clinical-like PDAC glandular structures were formed inside the model, providing a platform for simple simulation of stromal physical effects on PDAC cells. To investigate the pathogenesis of collagen-based tumor tissues, we examined transcriptome changes in 2D and collagen-based 3D PDAC cultures. Furthermore, to comprehensively explore the type I collagen-induced physical effects at advanced stages of PDAC, we performed metabolomic analysis to identify key metabolites that influence biological pathways. Our findings suggest that type I collagen and its physical effects are crucial factors in the development of PDAC and that this 3D PDAC collagen model has significant implications for understanding the intricate mechanisms involving type I collagen and PDAC.

## 2. Materials and Methods

### 2.1. Cell and 2D Culturing Conditions

The human pancreatic cancer cell line Capan-1 (ATCC HTB-79) was cultured as per the previous protocol [17,18]. Briefly, Capan-1 cells were maintained in Dulbecco’s Modified Eagle’s medium (DMEM; FUJIFILM Wako Pure Chemical Corp., Osaka, Japan) supplemented with 20% fetal bovine serum (FBS), 1% nonessential amino acids, and 1% streptomycin–penicillin at 37 °C in a humidified atmosphere containing 5% CO_2_.

### 2.2. Preparation of PDAC-Embedded 3D Collagen Tissue Matrix

To establish the collagen-based 3D PDAC model, the Collagen–Capan-1 model, PDAC cells (1 × 10^5^/mL) were suspended in neutralized type I collagen solution (IAC-30; Koken Co., Tokyo, Japan). The collagen mixture was poured into tissue culture plates pre-coated with 2-methacryloyloxyethyl phosphorylcholine (MPC) polymer (LIPIDURE-CM5206; NOF Co., Tokyo, Japan). The volume of collagen mixture was 800 μL/well when using 24-well plates and 200 μL/well when using 96-well plates. The corresponding volume of collagen solution was added to prepare collagen-only controls. Then, the plates were incubated at 37 °C for 30 min for gelation, and the gels were released from the wells using a vortex mixer. Additional fresh medium of 1000 μL and 100 μL was added to 24- and 96-well plates, respectively. The plates were placed in a 37 °C incubator with shaking at 95 rpm for extended culture. The medium was changed every 2–3 days and supplemented with 10% FBS, 1% nonessential amino acids, and 1% streptomycin–penicillin. Gel contraction was calculated as the percentage of lattice area relative to the initial gel area using ImageJ (1.53t Java 1.8.0_322, U.S. National Institutes of Health, Bethesda, MD, USA) software.

### 2.3. RNA Sequencing Analysis

RNA was extracted from Capan-1 cells that were normally cultured in dishes, referred to as the 2D Capan-1 cells, using NucleoSpin RNA (Macherey Nagel GmbH & Co., KG, Duren, Germany) and from the Collagen–Capan-1 model after 3 h of incubation using ISOGEN reagent (Nippon Gene; Tokyo, Japan), following the manufacturer’s instructions. RNA sequencing analysis of 2D Capan-1 and Collagen–Capan-1 was performed as previously described [19]. To elucidate the functions of differentially expressed genes, Gene Ontology (GO) enrichment analysis was performed using the online database for annotation, visualization, and integrated discovery (DAVID). Furthermore, Gene Set Enrichment Analysis (GSEA) was performed using GSEA software (GSEA software 4.2.3, Broad Institute, Cambridge, MA, USA). The raw sequences in FASTQ format are available from DDBJ (DRA017362).

### 2.4. Histological Analysis

To preserve the collagen matrix structure, the Collagen–Capan-1 model was encapsulated using iPGell (Genostaff, Tokyo, Japan). Paraffin-embedded sections with a thickness of 5 μm were then prepared. Hematoxylin and eosin (H&E) staining was performed for internal morphological analysis according to standard protocols.

### 2.5. Subcutaneous Transplantation In Vivo

Seven-week-old female nude mice (BALB/c-nu/nu; CLEA Japan, Tokyo, Japan) were maintained at the Animal Center of the National Institute of Advanced Industrial Science and Technology (AIST). The Collagen–Capan-1 model was prepared in 24-well plates and harvested on Day 7. Subsequently, Matrigel with three models was gelated to integrate one tumor fragment at room temperature. This fragment was subsequently transplanted into the left thigh of mice (n = 3). On Day 49, the mice were euthanized by cervical dislocation, and all subcutaneous tumors were isolated. All invasive procedures were conducted under isoflurane anesthesia, following guidelines approved by the Institutional Animal Care and Use Committee of the respective institutes of AIST.

### 2.6. Metabolomic Analysis

Primary metabolites were analyzed using liquid chromatography–mass spectrometry (LC-MS). Briefly, metabolites were extracted by vortexing the culture medium with ice-cold methanol. Subsequently, the metabolites were purified and analyzed by LC-MS according to the specific experimental methodology of a previous study [20].

### 2.7. Statistical Analysis and Visualization

The dataset containing metabolite information was converted to CSV format and then uploaded onto the MetaboAnalyst^®^ platform (accessible at [https://www.metaboanalyst.ca] accessed on 2 November 2023). This platform is equipped to perform extensive processing and analysis of metabolomic data. To ensure data integrity, a default data integrity check was performed, following a filtration process based on the average intensity values. A significance level was set at *p* < 0.05, and the Pareto data scaling was used for normalization when comparing two group samples. Orthogonal-partial-least-squares discrimination (OPLS-DA) was conducted. The OPLS-DA results were visualized using principal component analysis (PCA) and heatmap cluster analyses. Additionally, the visualization of variable important in projection (VIP) plots was employed following the OPLS-DA results to pinpoint discriminative metabolite markers.

## 3. Results

### 3.1. Construction of PDAC Cell-Embedded 3D Collagen Tissue Matrix

Capan-1 cells were embedded in a type I collagen gel to construct a collagen-based 3D PDAC model (Figure 1A), referred to as the Collagen–Capan-1 model. This model was used to investigate the effects of type I collagen on the progression of PDAC. As shown in Figure 1B, the type I collagen gel with no PDAC cells acted as a control group (collagen-only); the Collagen–Capan-1 model presented a disk structure on Days 7 and 14, in contrast to the loose structure of the collagen-only group, suggesting that interactions between type I collagen and Capan-1 cells occurred within the model.

### 3.2. Effect of Type I Collagen-Derived Physical Stress on PDAC Cells

To understand the impact of type I collagen-derived physical stress on Capan-1 cells during the initial modeling stages, RNA sequencing analysis was used to evaluate alterations in global gene expression patterns of Capan-1 cells following co-culture with type I collagen for 3 h. Compared to Capan-1 cells that were cultured in dishes, referred to as the 2D Capan-1 cells, 1103 genes were differentially expressed in the transcriptome of the Collagen–Capan-1 model, of which 597 genes were upregulated, and 606 were downregulated (Figure 2A). Representative upregulated (*FGF1*, *MYC*, and *IL6*) and downregulated genes (*SEMA4D*, *PCDH20*, and *ISLR*) are shown in Figure 2A. In addition, detailed upregulated genes associated with tumor proliferation are shown in Figure 2B. On the other hand, downregulated genes associated with reduced cell adhesion, signaling increased metastasis and migration of PDAC cells, are shown in Figure 2C.

To further explore alterations involving biological functions, upregulated and downregulated genes with significant differences were subjected to Gene Ontology (GO) analysis, and the top five biological processes (BP) were listed. Notably, although the same BPs were listed in both up and down situations, the content gene sets were different. The representative GO terms related to upregulated genes were “positive regulation of cell proliferation “and “positive regulation of gene expression” (Figure 3A, Appendix A), and that related to downregulated genes was “cell adhesion” (Figure 3B, Appendix A), indicating the crucial role of type I collagen in proliferation and metastasis of PDAC cells. In addition, GSEA analysis was performed to identify pathways enriched in the ranked gene lists. The results demonstrated that actin-related pathways were enriched at the initial modeling stages, such as “Actin polymerization or depolymerization”, “Actin mediated cell contraction”, and “Actomyosin structure organization” (Figure 3C–E).

### 3.3. Internal Histological Analysis of Collagen–Capan-1 Model

As depicted in Figure 4A, B, H&E staining was used to identify the internal morphology of the Collagen–Capan-1 model. Compared with the cell-free model (collagen only), the Collagen–Capan-1 model displayed clinic-like glandular structures of PDAC on days 7 and 14. Notably, the Collagen–Capan-1 model exhibited a more mature glandular structure on Day 14, which potentially resembled an advanced stage of PDAC. Notably, these ductal structures were more abundant than those in 2D-cultured Capan-1 cells, which showed no such ducted glandular structures (Figure 4C) and recapitulated the glandular structures that were built in tumor subcutaneous transplantations in the in vivo model (Figure 4D). These findings highlight the significant role of collagen 3D matrix in supporting glandular structure formation in PDAC.

### 3.4. Identifying Secreted and Consumed Extracellular Metabolites in Collagen–Capan-1 Model

To investigate the role of type I collagen-induced physical effects on the metabolic mechanism of Capan-1 in the 3D matrix, metabolomic analysis of water-soluble metabolites collected from culture supernatants was performed using LC-MS. OPLS-DA statistical analysis was utilized to determine differential metabolic patterns in the Collagen–Capan-1 group compared to the 2D Capan-1 group (Figure 5A). In addition, a heatmap was generated to provide a detailed understanding of water-soluble metabolites, highlighting the top 25 metabolites with significant variation (Figure 5B). Visualization of the VIP plot also emphasized distinct metabolites between the two groups (Figure 5C). In detail, in the culture supernatant of the Collagen–Capan-1 model, a higher accumulation of glutamic acid and lactic acid was detected (Figure 5D), indicating a metabolic shift of Capan-1 cells toward glycolysis strongly in the 3D collagen matrix [21,22]. Interestingly, proline, known to support PDAC growth under hypoxic conditions as a nutrient, also exhibited elevated levels. This suggests that collagen has the potential to provide a hypoxic tumor microenvironment that is conducive to PDAC cell growth. Consumption of essential nutrients such as lysine and glutamine (Figure 5E) was also observed in Collagen–Capan-1 supernatants. Collectively, these findings suggest that PDAC cells can adapt to the metabolic mechanisms of glycolysis in a 3D collagen matrix, resembling clinical PDAC metabolism. These findings underscore the vital role of type I collagen in the context of PDAC.

## 4. Discussion

In this study, we established a type I collagen-based 3D PDAC model, offering insights into the physical effects of type I collagen on the pathogenesis and progression of PDAC. Histologically, a clinical-like glandular structure of PDAC was spatially developed in the 3D collagen matrix, facilitating a realistic understanding of interactions between PDAC cells and the interplay of collagen on PDAC cells. Transcriptomic analysis was employed to elucidate the potential of type I collagen-induced physical effects in promoting tumor proliferation and migration during the early modeling stage, mimicking the malignant role of elevated type I collagen in the pathogenic stage of PDAC. Furthermore, metabolomic analysis revealed that the type I collagen-based 3D matrix induced a metabolic shift toward a glycolytic phenotype in PDAC cells. Additionally, it provides proline to support the rapid proliferation of PDAC cells under hypoxic and nutrient-depleted conditions.

In GSEA analysis, actin-related biological properties were activated at the initial modeling stages, suggesting an interplay between collagen and PDAC cells [23,24]. Actin, an integral component of the cytoskeleton, is responsible for the maintenance and modification of cellular morphology [24,25]. Specifically, the dynamic reorganization of actin occurs under the influence of collagen, leading to a potential enhancement in cellular contraction, metastasis, and invasion by PDAC cells, which aligns with the reduction in “cell adhesion” in the GO analysis and the reduced expression of associated adhesion related molecule genes such as *SEMA4D*, *PCDH20*, and *ISLR* [26,27,28]. Moreover, Capan-1 cells may adjust their mechanical properties to adapt to the surrounding microenvironment, which is altered by collagen-induced mechanical stress via actin activation. This explains the positive relationship between mechanical stress in the tumor microenvironment and the capacity for proliferation and invasion by PDAC cells [29]. Collectively, these results provide a theoretical basis for collagen as a potential therapeutic target in PDAC.

PDAC cells, like other epithelial cancer cells, require substantial amounts of energy to support rapid proliferation. To meet this demand, PDAC cells can shift their metabolic pathways based on different tumor microenvironments. The adaptability of PDAC cells is a critical factor that contributes to the difficulty of PDAC treatment [30]. Alterations in the metabolic processes of Capan-1 cells between the 2D culture and the 3D collagen matrix also confirmed this hypothesis. In the Collagen–Capan-1 culture medium, a notable accumulation of lactic acid and glutamic acid was observed, suggesting that the high proliferation of Capan-1 cells relies on the glycolytic pathway, commonly known as the Warburg effect [31,32]. The excessive consumption of nutrients like lysine, valine, and histidine in the Collagen–Capan-1 medium, compared to the 2D Capan-1 culture, suggests a promotional effect of mechanical stress on the proliferation of Capan-1 cells. Moreover, increased proline expression implies that the physical effect and type I collagen can serve as a proline source to maintain PDAC cell proliferation under nutrient-depleted and hypoxic conditions [33,34]. Proline is catalyzed by proline dehydrogenase 1 (PEODH1) to generate glutamine, which is involved in the tricarboxylic acid (TCA) cycle for ATP production [20,35,36]. Furthermore, the notable reduction in tryptophan in Collagen–Capan-1 medium potentially indicates the activation of the tryptophan metabolic pathway in PDAC cells, implying that the mechanical stress of the 3D collagen matrix potentially supports the generation of immune evasion in this 3D PDAC model. Taken together, these findings demonstrate that PDAC cells employ a unique survival strategy in a 3D stromal environment, highlighting the matrix’s supportive role in PDAC progression.

Type I collagen is a primary component of ECM that contributes to drug resistance in PDAC [37]. For instance, it potentiates drug resistance to gemcitabine in pancreatic cancer cells [13,14]. First, type I collagen forms a physical barrier surrounding PDAC, impeding the delivery and efficacy of chemotherapeutic agents [38,39]. Additionally, type I collagen can lead to the development of PDAC resistance via interactions with cell-surface receptors and intercellular signaling pathways. Membrane-type 1-MMP (MT1-MMP) plays a central role in collagen-mediated drug resistance in PDAC [40]. Overexpression of MT1-MMP in the collagen microenvironment increases ERK1/2 phosphorylation and high-mobility group AT-hook 2 (HMGA2) expression, which further attenuates GEM-induced checkpoint arrest by acting as a means of drug resistance [41,42]. Consequently, type I collagen holds promise as a potential therapeutic target in PDAC. At the molecular level, a thorough understanding of the role of type I collagen in PDAC progression will be beneficial for future research focused on collagen-based drug resistance.

In this study, we conducted an in-depth exploration of type I collagen in PDAC, which is predominant in the ECM, to investigate its physical effects on PDAC cell proliferation. However, the absence of other cellular and extracellular matrix components in the Collagen–Capan-1 model makes it difficult to apply it to drug screening and research concerning disease mechanisms that require a higher demand for the simulation of PDAC microenvironments. Specifically, for instance, many cell types are presented in the PDAC stromal environment, including cancer-associated fibroblasts, immune cells, and epithelial cells. In addition, a diverse array of ECM components is included, comprising fibronectin, hyaluronic acid, laminin, matrix metalloproteinases [43], and multiple types of collagens, such as type III [44] and type IV [45]. Each of these components plays a significant role in PDAC progression and chemoresistance [46,47], and the incorporation of additional components into our 3D model offers the potential to simulate a more authentic PDAC stromal microenvironment. Therefore, to achieve a more realistic PDAC microenvironment, it is imperative to consider the inclusion of a wide range of PDAC components in our next research phase.

## 5. Conclusions

In the present study, we constructed a collagen-based 3D PDAC model with clinical glandular structures using Capan-1 cells. The physical role of type I collagen in promoting PDAC proliferation and migration was confirmed using molecular studies of global gene expression patterns and metabolomic analyses. These findings enhance our understanding of how collagen and mechanical stress, influenced by the stromal environment, affect the clinical PDAC microenvironment. This, in turn, facilitates further exploration into the remodeling of the PDAC microenvironment.

## Figures and Tables

**Figure 1 bioengineering-10-01437-f001:**
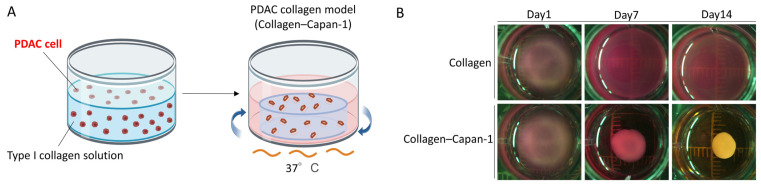
Construction of PDAC-embedded 3D collagen tissue matrix. (**A**) Graphical representation of PDAC collagen model establishment. (**B**) Physical characteristics of PDAC collagen model. The Collagen-only model was used as a control. Both groups were modeled in 24-well plates and observed on Days 1, 7, and 14.

**Figure 2 bioengineering-10-01437-f002:**
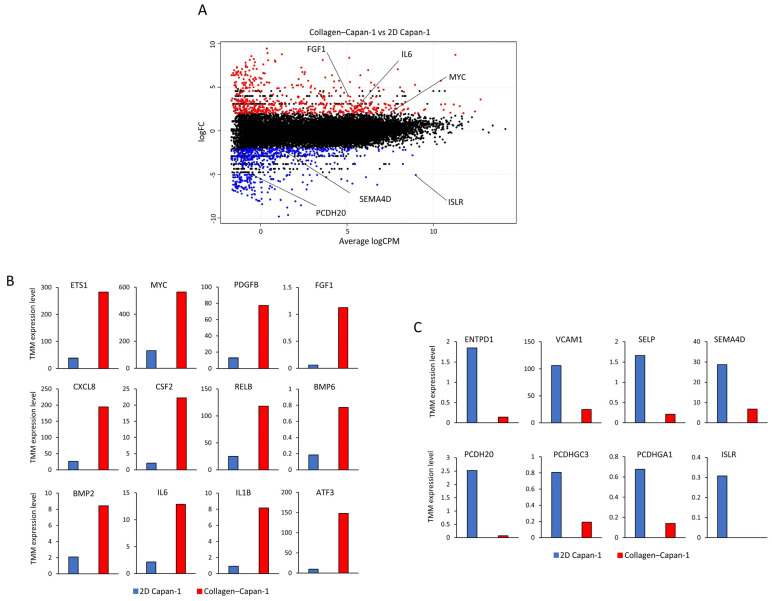
Differential gene expression analysis of PDAC cells triggered by collagen 3D culturing at the initial modeling phase, as revealed by RNA-seq analysis (n = 1, respectively). (**A**) The moving average (MA) plot displays differentially expressed genes in PDAC cells following 3 h culture in collagen 3D matrix. Expressed genes with significant differences were adjusted by *p* < 0.05 and log2 [fold-change] ≥ 2. The gene expression of PDAC cells in 2D culture as a basis. Upregulated genes are represented in red (n = 597), while downregulated genes are represented in blue (n = 606). (**B**) Representative upregulated markers associated with cell proliferation in the Collagen–Capan-1 model. (**C**) Representative downregulated markers associated with cell adhesion in the Collagen–Capan-1 model. The trimmed mean of M values (TMM) expression level in Collagen-Capan-1 is represented by red bars, and the corresponding TMM expression in 2D Capan-1 is represented by blue bars.

**Figure 3 bioengineering-10-01437-f003:**
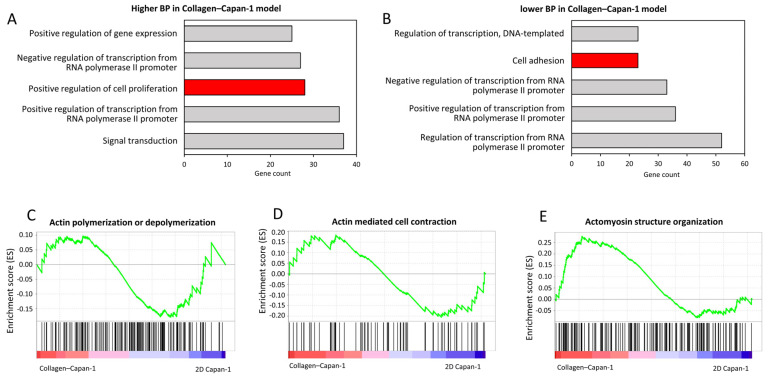
Alterations in global gene expression patterns of PDAC cells identified via RNA-seq analysis after 3 h incubation with type I collagen (n = 1, respectively). (**A**) GO terms (biological process) enriched in the Collagen–Capan-1 model. The red bar indicates GO terms related to promotional tumor functions, such as cell proliferation. (**B**) GO terms (biological process) reduced in the Collagen–Capan-1 model. The red bar indicates GO terms related to tumor migration and metastasis functions, such as cell adhesion. (**C**) GSEA of Collagen–Capan-1 model, compared to 2D Capan-1 culture. Enrichment plots of expression signatures of “Actin polymerization or depolymerization”, (**D**) “Action mediated cell contraction”, and (**E**) “Actomyosin structure organization”.

**Figure 4 bioengineering-10-01437-f004:**
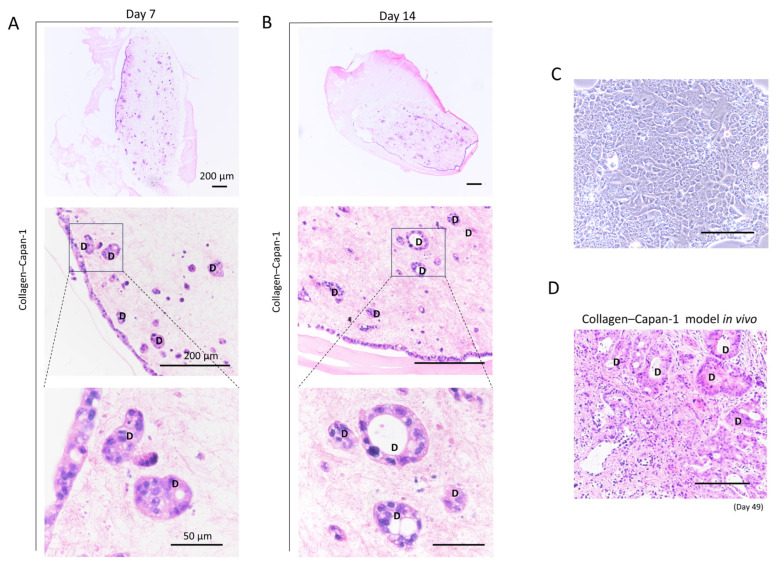
Internal histological analysis of 3D collagen PDAC model. (**A**) Representative images of H&E staining in Collagen–Capan-1 model on Day 7. (**B**) Representative images of H&E staining in Collagen–Capan-1 model on Day 14. D indicates duct formation. Scale bars = 200 μm, 200 μm, and 50 μm, respectively. (**C**) Morphology of Capan-1 cells in 2D culture. Scale bar, 200 μm. (**D**) Histological analysis of H&E-stained tumor cells generated subcutaneously in vivo in the Collagen–Capan-1 model. D indicates duct formation. Scale bar = 100 μm.

**Figure 5 bioengineering-10-01437-f005:**
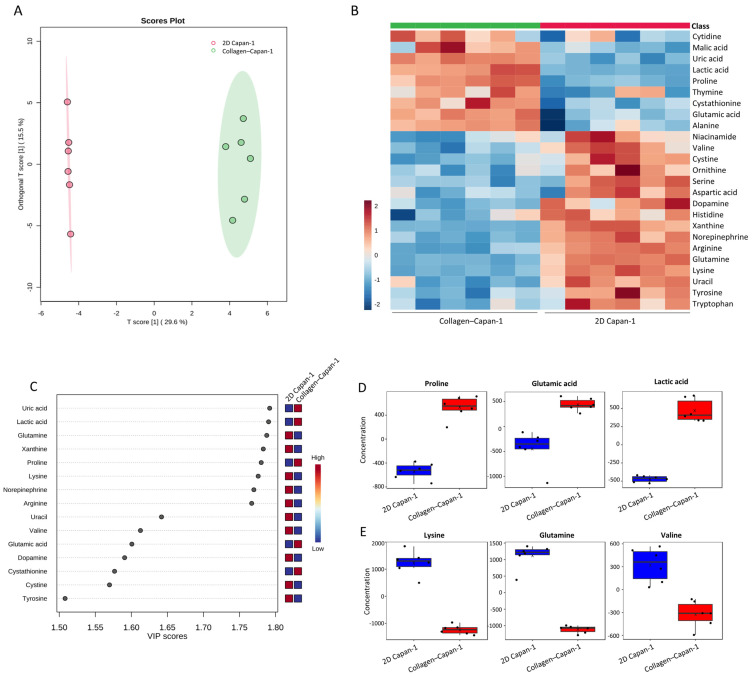
Metabolic alterations to Capan-1 cells in 3D collagen matrix. (**A**) Different metabolites detected in the supernatants of Capan-1 cells in 2D culture and collagen 3D matrix. Metabolomic datasets were subjected to OPLS-DA analysis using MetaboAnalyst 5.0. The OPLS-DA score plot visually distinguishes between two sample groups (n = 6 samples per group) based on metabolomic data: the supernatant of Capan-1 cell in 2D culture is represented in red, while the supernatant of Capan-1 cells in collagen 3D matrix is represented in green. The model consists of one predictive x-score component: component t [1] and one orthogonal x-score component to [1]. t [1] explains 29.6% of the predictive variation in x, and to [1] explains 15.5% of the orthogonal variation in x. (**B**) Heatmap comparing metabolites changed in the supernatants of Capan-1 cells in 2D culture and collagen 3D matrix. (**C**) VIP plot corresponding to the score plot of OPLS-DA to visualize metabolite markers that contributed to the discrimination between the supernatants of Capan-1 cells in 2D culture and collagen 3D matrix. (**D**) Representative upregulated metabolites (proline, glutamic acid, and lactic acid) in the supernatants of Capan-1 cells cultured in collagen 3D matrix. (**E**) Representative downregulated metabolites (lysine, glutamine, valine) in the supernatants of Capan-1 cells in collagen 3D matrix culture. The metabolite concentration of Capan-1 collagen is represented by the red bar, while the metabolite concentration of 2D Capan-1 is represented by the blue bar.

## Data Availability

All other data supporting the findings of this study are available from the corresponding author upon request.

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
