# Peer review of "Exploring the Role of Desmoplastic Physical Stroma in Pancreatic Cancer Progression Using a Three-Dimensional Collagen Matrix Model"

_bioengineering, 2023, doi:10.3390/bioengineering10121437_

Round 1

Reviewer 1 Report

Comments and Suggestions for Authors

The authors present a 3D cell culture model to study the progression of pancreatic cancer progression. Using collagen and nutrient based 3D models they present the formation of tumoroids which can be further characterized for their progression. Through mass-spec they identify key markers in the progression of this cancer namely Capan1 and other factors which can contribute significantly to the disease progression.

In the current study, the authors have identified several factors which is most common in most cancer cells. However, it would have been good to dissect 1-2 proteins in more details to show their mechanistic roles in cancer progression. 

This point is missing and should be discussed in discussion and conclusion section, Also the current conclusion section is not satisfactory and should clearly reflect the key findings of the study, shortcomings and future directions of the study. That will make the paper complete. 

Also, in figure 2 and 3 graphs, I dont see an error bars. Is it just one experiment or average of multiple. The authors should clearly mention it and should represent the error bars and statistical significance wherever necessary. 

Comments on the Quality of English Language

The English language is properly used. However, I suggest authors to get it checked by software such as Grammarly for proper usage of terms and adjectives. 

Author Response

Thank you for your insightful feedback on our manuscript. We have revised our manuscript in light of your suggestions, with a focus on enhancing the discussion and conclusion sections.

Discussion Section Revision:

In response to your suggestion, we have included a more detailed analysis of tryptophan in the context of our 3D collagen matrix model for pancreatic ductal adenocarcinoma (PDAC). The revised manuscript now discusses how the reduction of tryptophan potentially indicates the activation of the tryptophan metabolic pathway in PDAC cells, suggesting that mechanical stress from the 3D collagen matrix supports immune evasion in this model (L304-307).

Conclusion Section Revision:

We have also refined the conclusion to better reflect the key findings, shortcomings, and future directions of our study. The revised conclusion emphasizes the role of type I collagen in promoting PDAC proliferation and migration and its implications in the clinical PDAC microenvironment (L341-347).

Clarification on Figures 2 and 3:

Regarding your concern about the absence of error bars in Figures 2 and 3, we clarify that the data presented are from a one-time RNA-seq analysis, focusing on TPM values from individual samples. We understand the importance of this clarification and have marked (n=1) in the legends of these figures (L176, L199). While each data point represents a single sample, the depth of RNA-seq data allows for meaningful comparisons in this context.

We hope these revisions and clarifications adequately address your comments and improve the manuscript. We look forward to your further guidance and feedback.

The manuscript has been revised with new sentences marked in red. Please check the attached new version of the manuscript.

Reviewer 2 Report

Comments and Suggestions for Authors

This study established a 3D Collagen Matrix Model to examine type I collagen-induced physical stress effects on Capan-1 cells (a human pancreatic ductal adenocarcinoma (PDAC) cell line) proliferation and migration. Using transcriptome analysis, a link between type I collagen-induced physical effects and promotion of Capan-1 cell proliferation (e.g., upregulated genes FGF1, MYC, and IL6) and migration (e.g., downregulated genes SEMA4D, PCDH20, and ISLR) was determined. Gene Ontology analysis and GSEA analysis of upregulated genes and downregulated genes indicated the crucial role of type collagen in proliferation and metastasis of Capan-1 cells. Internal histological analysis of Collagen-Capan-1 model in vitro and in vivo (subcutaneous transplantation to nude mice) highlighted the significant role of collagen 3D matrix in supporting glandular structure formation in PDAC. Metabolomic analysis revealed that the type I collagen-based 3D matrix induced a metabolic shift toward a glycolytic phenotype in PDAC cells and it provides proline to support the rapid proliferation of PDAC cells under hypoxic and nutrient-depleted conditions. All these findings are consistent with previous reported studies. The incorporation of additional components into the present 3D model will offer the potential to simulate a more real PDAC stromal microenvironment. In addition, it is worth using the present 3D Collagen Matrix Model to examine the notion that extracellular matrix collagen I differentially regulates the metabolic plasticity of PDAC parenchymal cell and cancer stem cell (Cancers (Basel). 2023 Jul 29;15(15):3868.  doi:10.3390/cancers15153868). There are some minor concerns as listed in the following:

*L38: PDAC: full name first

*L49: type I collagen: give the source

*L124: This fragment was subsequently transplanted into the mice: give the transplanted location

L141: average intensity values. p < 0.05. difference.

L146: markers.3. Results

*L280: cells adjust may their mechanical properties

@ References: Inconsistent writing format for the Journal name

L397: The FEBS Journal -> FEBS J.

L413: Journal of Hematology & Oncology -> J Hematol Oncol

L416: Proceedings of the National Academy of Sciences

L425: Molecular Cancer -> Mol Cancer

L431: Frontiers in Cell and Developmental Biology

L433: Biochimica et Biophysica Acta (BBA) - Reviews on Cancer -> Biochim Biophys Acta Rev Cancer

L470: Journal of Translational Medicine

Comments on the Quality of English Language

Minor editing of English language required

Author Response

Thank you for your constructive feedback on our manuscript. We have made the following revisions based on your suggestions:

Terminology Clarifications:

L38: We have clarified the full name of PDAC (Pancreatic ductal adenocarcinoma) at its first mention.

L49: The source of type I collagen has been added in L89 (IAC-30; Koken Co., Tokyo, Japan).

Methodological Details:

L124: We have specified the transplantation location in L126, stating that the fragment was transplanted into the left thigh of mice (n = 3).

Statistical and Analytical Clarifications:

L141: Revised L142 to include a clear mention of the significance level (p < 0.05) and the use of Pareto data scaling for normalization in comparative analyses.

Text Corrections:

L280: Corrected the sentence structure in L283 to "cells may adjust their mechanical properties."

Reference Formatting:

We have revised the journal names in the references to maintain consistency in the citation format, as per your specific instructions. Corrections include changes to references at L405, L421, L433, and L441. For other cited journals (L416 -> L423, L431-> L438, L470->L477), we have retained the current format as it is correct.

We believe these revisions enhance the clarity and accuracy of our manuscript. We appreciate your guidance in making these improvements and hope the manuscript now meets your expectations.

The manuscript has been revised with new sentences marked in red. Please check the attached new version of the manuscript.
